# Precision Nutrition Opportunities to Help Mitigate Nutrition and Health Challenges in Low- and Middle-Income Countries: An Expert Opinion Survey

**DOI:** 10.3390/nu15143247

**Published:** 2023-07-21

**Authors:** Jacquelyn R. Bedsaul-Fryer, Kesso G. van Zutphen-Küffer, Jimena Monroy-Gomez, Diane E. Clayton, Breda Gavin-Smith, Céline Worth, Christian Nils Schwab, Mathilda Freymond, Anna Surowska, Laís Bhering Martins, Christina Senn-Jakobsen, Klaus Kraemer

**Affiliations:** 1*Sight and Life*, P.O. Box 2116, 4002 Basel, Switzerland; kesso.vanzutphen@sightandlife.org (K.G.v.Z.-K.); jimena.monroy@sightandlife.org (J.M.-G.); mathilda_freymond@sightandlife.org (M.F.); 2Department of Human Nutrition & Health, Wageningen University & Research, 6708 PB Wageningen, The Netherlands; 3York Consumer Health, Route Du Charmin 15, 1648 Hauteville, Switzerland; dianeclayton@bluewin.ch; 4Nestlé, Corporate R&D, Av. Nestlé 55, 1800 Vevey, Switzerland; celine.worth@nestle.com; 5Integrative Food and Nutrition Center, École Polytechnique Fédérale de Lausanne, Rte Cantonale, 1015 Lausanne, Switzerland; christian.schwab@epfl.ch; 6EssentialTech Centre, École Polytechnique Fédérale de Lausanne, Rte Cantonale, 1015 Lausanne, Switzerland; anna.surowska@gmail.com; 7Swiss Food & Nutrition Valley, EPFL Innovation Park, Station 12, 1015 Lausanne, Switzerland; lais@sfnv.ch (L.B.M.); christina@sfnv.ch (C.S.-J.); 8Department of International Health, Johns Hopkins Bloomberg School of Public Health, Baltimore, MD 21218, USA

**Keywords:** precision nutrition, precision public health nutrition, LMIC, malnutrition

## Abstract

Precision nutrition involves several data collection methods and tools that aim to better inform nutritional recommendations and improve dietary intake, nutritional status, and health outcomes. While the benefits of collecting precise data and designing well-informed interventions are vast, it is presently unclear whether precision nutrition is a relevant approach for tackling nutrition challenges facing populations in low- and middle-income countries (LMIC), considering infrastructure, affordability, and accessibility of approaches. The Swiss Food & Nutrition Valley (SFNV) Precision Nutrition for LMIC project working group assessed the relevance of precision nutrition for LMIC by first conducting an expert opinion survey and then hosting a workshop with nutrition leaders who live or work in LMIC. The experts were interviewed to discuss four topics: nutritional problems, current solutions, precision nutrition, and collaboration. Furthermore, the SFNV Precision Nutrition for LMIC Virtual Workshop gathered a wider group of nutrition leaders to further discuss precision nutrition relevance and opportunities. Our study revealed that precision public health nutrition, which has a clear focus on the stratification of at-risk groups, may offer relevant support for nutrition and health issues in LMIC. However, funding, affordability, resources, awareness, training, suitable tools, and safety are essential prerequisites for implementation and to equitably address nutrition challenges in low-resource communities.

## 1. Introduction

Recent technological innovations have catalyzed a shift in human nutrition research and interventions toward a concept in nutrition science referred to as “personalized nutrition” [1,2], “precision nutrition” [3,4,5,6,7,8,9], “targeted nutrition” [10,11], and “nutrition for precision health” [12]. While distinct, these terms share common core properties, such as tailoring nutritional recommendations and solutions to account for variations related to phenotype, genotype, other biological features, environment, social factors, lifestyle, behavior, health goals, and preferences. Precision nutrition is an encompassing term for an integrative methodology that includes a variety of approaches used to gather general and detailed data (Figure 1) with the goal of maintaining or improving nutritional status and health [1,2,3,4,5,6,7,8,9,10,11,12].

The holistic methodology of precision nutrition includes platforms for data collection, such as demographic surveys and lifestyle questionnaires that are already widely used in public health nutrition and provide general data, as well as advanced tools and methods. Such advanced approaches include mobile data collection and quantification [13,14], metabolic indicator assessments, and genetic and omics-based methods, which have high specificity but also high technical requirements, and are largely inaccessible to consumers and practitioners in low-resource settings [7]. The dynamic data collection methods of precision nutrition with wider accessibility and the advanced approaches provide comprehensive information when implemented together and support the development of tailored, well-informed nutritional recommendations and solutions [1,2,3,4,5,6,7,8,9,10,11,12,13,14].

Currently, the precision nutrition tools and methods with enhanced specificity remain largely confined to research settings in high-income countries (HIC) [7,15], where funding, training, and access pose fewer challenges [16,17,18,19]. Implementation of these specific precision nutrition methods is primarily focused on individuals in HIC, where it is more feasible to analyze genetics, metabolism, and the microbiome, among other characteristics, at the individual level for personal benefits [15,20,21,22,23]. Furthermore, most of the data collected to inform nutritional recommendations come from HIC; for example, nearly 70% of the human microbiome data from the three largest genomic data repositories were collected from individuals in the United States, Europe, and Canada, while central and southern Asia was the most underrepresented region (e.g., Bangladesh and India represented only 0.8% and 0.7%, respectively) [15]. However, limiting precision nutrition applications to individuals in HIC will inherently lead to limitations in research and missed opportunities to address nutrition and health challenges in low-and middle-income countries (LMIC).

Today, approximately three billion people in the world cannot afford a healthy diet [24], and 45% of deaths among children under five years of age are linked to undernutrition, primarily in LMIC [25]. In addition, it is expected that 670 million people will face hunger by 2030 [24]. While traditional one-size-fits-all approaches in public health nutrition have been efficacious [26,27], limitations in this strategy have caused unintended harm in certain instances for certain subgroups [28,29]. Thus, well-informed nutritional interventions that are tailored for at-risk subgroups within a target population are urgently needed to combat malnutrition and its associated health issues, and to prevent unintended harm [30].

The concept of precision nutrition for LMIC populations is emerging as a promising component of the solution to mitigate malnutrition and improve health by enhancing the accuracy in data collection and analysis, providing detailed information about the biology and other features of subgroups for targeted interventions [2,3,31,32,33]. The accurate information collected supports the development of safer, more efficacious interventions, although methods are not widely practiced nor available globally.

The Swiss Food & Nutrition Valley (SFNV) Precision Nutrition for LMIC project, established in 2021 and led by *Sight and Life*, aims to make precision nutrition more accessible, affordable, and desirable to low-resource communities as an approach to address major nutritional challenges facing these regions. However, it is not fully known whether the concept and application of precision nutrition are considered relevant for LMIC. Consequently, the working group sought to answer this question by developing an expert opinion survey and subsequently hosting a workshop with nutrition scientists, practitioners, and professionals who live in or have direct working experience in LMIC. These efforts aimed to build a first-hand and deeper understanding of the needs, challenges, and opportunities that precision nutrition can offer for the benefit of populations in low-resource settings.

This article disseminates the novel insights gained through expert opinions regarding the region-specific nutritional challenges; current methods and solutions; precision nutrition relevance and opportunities; collaboration; considerations for developing projects; and the way forward from the research and implementation perspectives.

## 2. Materials and Methods

This study includes both qualitative and quantitative research methods to survey the relevance of precision nutrition to address nutrition and health challenges in LMIC. A literature review was conducted to identify any current applications or examples of precision nutrition in low-resource settings. Nutrition leaders who live or work in LMIC, identified from the working group network and the literature review, were invited to share opinions on precision nutrition for LMIC via an interview and online survey.

This protocol was not previously published. It represents an adaptation of the Centers for Disease Control (CDC) guidelines for conducting a needs assessment [34,35,36,37,38]. The guidelines considered fall into the following categories: planning, question development, data collection, and data analysis. In addition, we used similar guiding questions to host a workshop to disseminate key findings of the expert opinion survey and to further identify precision nutrition relevance and opportunities for LMIC.

### 2.1. Expert Opinion Survey

#### 2.1.1. Planning

The working group considered five key points listed in the CDC guidelines [34,35,36,37,38] during the survey planning:What is the primary purpose of the assessment?What existing data do we have?Who needs to participate so the results of the assessment are representative?How will we use the information?What resources are available (e.g., budget, people, and time)?

The planning procedure included five steps according to these guiding questions (Figure 2A). We defined the purpose of our work, and performed a literature review to identify any definitions, applications, approaches, tools, and examples of precision nutrition in LMIC contexts. The working group generated a list of experts for this survey based on nutrition experience in LMIC and expertise in the emerging precision nutrition field. The list included experts from various LMIC regions identified from the working group network and the literature review (Figure 2B). Eligibility criteria were that the experts must live or work in LMIC and hold a nutrition or public health position in academia, medicine, public health, government, industry, or civil society.

#### 2.1.2. Survey Question Development

To develop useful questions that aim to assess the relevance of precision nutrition for LMIC, the following four points of the CDC guidelines [34,35,36,37,38] were considered:Will the question yield useful information?Will the respondent be able to answer the question?Is the question necessary?What question format is best for gathering information (e.g., multiple-choice questions, open-ended questions, or checklists)?

The working group developed twenty-four questions on topics related to four themes: nutritional challenges, current methods and solutions, precision nutrition, and collaboration. The survey question design (Appendix A) facilitated open-ended answers to stimulate discussion.

#### 2.1.3. Data Collection

To determine the data collection methods, the following two points [34,35,36,37,38] were considered:What format best applies for data collection in this case?Will more than one format be utilized for data collection?

An online survey of questions was developed and emailed to the opinion leaders. Questions were accompanied by long answer sections to encourage open-ended written responses. The opinion leaders were not time- or space-restricted when providing answers. Submission of the online form was positioned as optional, but strongly encouraged. The working group scheduled an interview with each of the thirteen experts as the primary method for data collection. The survey questions were discussed during the interview that was conducted virtually with at least two working group members present. The interview agenda included introductions, a presentation on background concepts, and the interview that ranged from 45–75 min in total.

#### 2.1.4. Data Analysis

The online survey results were exported and analyzed via Microsoft Excel. The audio and video editing program Descript was used to transcribe the interview recordings into written transcripts. The survey results and transcripts were further analyzed, and data tables created via Microsoft Word. Themes were identified and aggregated from the qualitative data and quantitative measures were determined where feasible. Canva was used to visualize data through charts. The following CDC guiding questions from the planning stage above were considered when analyzing the data:What is the primary purpose of the assessment?How will you use the information?What resources are available to conduct a needs assessment (e.g., budget, people, and time)?

In addition, the content analysis research technique was adapted and implemented in this study to add quantitative measures. Specifically, summative content analysis was used to count and compare keywords and content from the interview and survey results. Next, the analysis was interpreted quantitatively using descriptive statistics [39,40]. An overview of the data was shared with the experts via a short article [7] and presentation at the workshop (Figure 2C).

### 2.2. Workshop

The Precision Nutrition for LMIC Virtual Workshop focused on the insights shared by LMIC nutrition experts who participated in the opinion survey and served as an opportunity to translate these needs into actionable steps. It also provided a platform for discussion with a wider network of opinion leaders in nutrition and technology on the use of precision nutrition approaches to help mitigate nutrition challenges in low-resource settings.

#### 2.2.1. Planning

The SFNV Precision Nutrition for LMIC project working group invited thirty-seven opinion leaders by email to the Precision Nutrition for LMIC Virtual Workshop based on suggestions by the experts of the opinion survey and the working group network. Selected opinion leaders worked in nutrition, public health, pediatrics, infectious disease, medicine, epidemiology, business, and technology, and lived or worked in LMIC or were colleagues of the working group and part of the Swiss ecosystem. Twenty-one opinion leaders and ten working group members attended the workshop. Prior to the workshop, all participants completed a short questionnaire to detail their interests and expertise related to precision nutrition and were assigned to brainstorming groups accordingly.

The workshop was organized into three parts: introductory session, brainstorming session, and closing session (Appendix A). The introductory session consisted of the background information on the SFNV Precision Nutrition for LMIC project, expert opinion survey learnings, and short presentations by LMIC nutrition experts on the nutritional challenges and precision nutrition opportunities to mitigate the challenges. All attendees were offered the opportunity to present their ideas.

Next, the brainstorming session facilitated small group reflections, which were moderated by a subject matter expert chosen by the working group based on his or her expertise in the topic and interest in leading the discussion. The topics discussed in each group were based on the precision nutrition approaches that were mentioned during the expert opinion survey and participant interest, as determined through the workshop questionnaire. The brainstorming discussion topics included:Precision nutrition—the collection and analysis of dietary intake and nutritional status data.Precision nutrition—nutritional products, supplements, and treatments.

The discussion questions (Appendix A) for the brainstorming session aligned with the brainstorming discussion topics above and were developed according to the guiding principles stated previously in methods Section 2.1.2.

Finally, the closing session consisted of each room’s discussion summary and feedback led by the moderators. The working group closed the workshop by providing input on the next steps toward action and impact.

#### 2.2.2. Analysis

The workshop data was analyzed via Descript and Microsoft Word as described in methods Section 2.1.4. The workshop recording and a post-workshop report were emailed to the attendees.

## 3. Results

### 3.1. Several Food and Nutrition Challenges Exist in LMIC

To identify prevalent food and nutritional challenges in different LMIC regions, the working group surveyed thirteen opinion leaders who live or work in sub-Saharan Africa, Southeast Asia, and South America. The specific countries represented in our analysis included: Côte d’Ivoire, Ghana, Malawi, The Gambia, South Africa, Bangladesh, India, Indonesia, Thailand, Vietnam, Brazil, and Ecuador.

The experts discussed food-specific challenges, including poverty, food insecurity, affordability of food, and the food production system (Table 1). In addition, the most stated nutritional challenge by the experts was undernutrition (Figure 3A), including protein-energy-malnutrition and micronutrient deficiencies, namely in zinc, iron, iodine, vitamins A, B_9_, and B_12_. These and other micronutrients were speculated to be lacking in LMIC populations [25], but the data were often insufficient to quantify the extent [41,42]. Phenotypic and health challenges associated with undernutrition were also discussed, including underweight, stunting, wasting, anemia, and acute illness such as infections [25]. Mortality and failure to thrive were described as severe outcomes from undernutrition (Table 1).

The opinion leaders also mentioned overnutrition and its related health challenges, such as hypertension, obesity, and other non-communicable diseases (NCDs) as prevalent issues in their regions (Table 1). The current barriers to addressing these food and nutrition challenges were discussed and further analyzed into four themes: awareness, resources, systems level, and collaboration barriers (Table 2).

Twelve experts (92.3%) classified nutrition awareness as a barrier to addressing these food, nutrition, and health challenges (Figure 3B). The barrier theme of awareness refers to individual consumers and constituents, public health and medical professionals, and politicians. Resources was another theme of barriers mentioned by ten experts (76.9%). Furthermore, nine experts (69. 2%) discussed systems-level barriers to addressing nutritional challenges, which spanned food security, poverty, organization of systems, accessibility, and availability of nutritious food. The remaining barriers were grouped into the collaboration theme. Eight opinion leaders (61.5%) discussed collaboration barriers among scientists, nutritionists, policy makers, and other key stakeholders (Figure 3B, Table 2).

Next, they were asked to suggest one target population or subgroup in need of an intervention against the aforementioned nutritional challenges. While it was agreed that all subgroups are important, children surfaced as the top group to target for intervention, with a total of five experts (38.5%) choosing this group (Figure 4). Of note, the group including preconception to children of two years of age was emphasized as a subgroup that should not be separated and, thus, was included as its own target group (23.1%). Adolescents (23.1%) and the elderly subgroups (15.4%) were also mentioned as top target groups for nutritional interventions (Figure 4).

### 3.2. Precision Nutrition for LMIC May Have Relevance for at-Risk Target Groups

#### 3.2.1. The Definition of Precision Nutrition for LMIC Must Emphasize the Targeting of at-Risk Groups

The working group supports the notion depicted in Figure 1 that precision nutrition includes widely practiced and accessible methods used in combination with the advanced, less accessible, specific methods that provide detailed data to develop improved nutritional recommendations and solutions. However, precision nutrition remains a loosely defined term in the literature, often used interchangeably with other terms such as personalized nutrition [7]. As an attempt to harmonize the understanding of this concept in the context of LMIC, the SFNV Precision Nutrition for LMIC project working group proposed a definition of precision nutrition: Precision nutrition is an approach that uses rigorous scientific information on an individual’s characteristics and environment. This information is used to develop targeted, accessible, affordable, desirable nutrition solutions that offer measurable individualized benefit. Such targeted solutions address the most pressing nutrition challenges in LMIC [7].

The experts were asked to critique this definition as part of the opinion survey. Their feedback was analyzed into two themes specific to term and definition-wide critiques. The term critiques focused on the words “individual”, “rigorous”, “LMIC”, and “affordability and accessibility”, while definition-wide critiques included concepts, such as its breadth, depth, and specificity (Table 3).

Overall, some experts believed the definition was broad but acknowledged it could become too bulky or specific if more depth were added. The stratification of subgroups as opposed to an individual focus was emphasized for the LMIC context. Together, these critiques led to a revision of the SFNV definition of precision nutrition for LMIC that is open for further critique. The updated, proposed SFNV definition of precision nutrition for LMIC is an approach that utilizes detailed scientific information on the biological, ecological, social, behavioral, and environmental factors of individuals or target groups to develop evidence-based, tailored, accessible, affordable, desirable, and safe nutrition solutions that offer measurable benefits and address the most pressing nutrition and health challenges in LMIC.

#### 3.2.2. Current Methods and Solutions to Address Nutritional Challenges Are Lacking

As part of the expert opinion survey, insights were shared regarding current solutions to the food and nutritional challenges in LMIC. For example, public health approaches such as point-of-care glucose screening, anthropometry, and food-based interventions were ongoing in some regions in this study (Table 4). In addition, many of the experts currently lead research programs that include precision nutrition methodology. Investigating the etiology of nutrition-related challenges, deep phenotyping, and implementation research (e.g., evaluation and programmatic) each can fall under the umbrella of precision nutrition approaches used to develop tailored recommendations and solutions.

While nutrition data were sparse and gaps remained [41,42], there are available resources for national data, including from agencies such as the World Health Organization, World Food Programme (WFP), governments, and other non-governmental organizations (NGOs) (Table 4).

The current methods described by the experts indicate that precision nutrition approaches may be possible in LMIC, primarily at the research level. Advanced tools listed in Table 4 (e.g., point-of-care devices for noncommunicable diseases) are in use, indicating feasibility in some instances. However, methods currently used more widely in practice provide less specificity and detail, leading to less precision in recommendations [1].

#### 3.2.3. Precision Nutrition Could Be Part of Nutrition Solutions for LMIC

All opinion leaders believed precision nutrition could be at least part of the solution and could find further relevance to help tackle the most pressing nutritional challenges in LMIC. However, they underscored that precision nutrition is “not just the collection of data” with tools and methods, but their “translation into an application” [7] (p. 59). In addition, the experts emphasized that ethical and cultural issues must be considered and addressed for precision nutrition to be a viable solution for LMIC [7].

The experts chose three precision nutrition approaches that could find further relevance to tackle nutrition challenges in their regions. The approaches discussed by the experts were analyzed into three categories: gene and omics-based tools (61.5%), microbiota- and gut health-based approaches (46.2%); and devices, apps, and digital health (38.5%) (Figure 5A). Omics-based tools consisted of nutrigenomics, genomics, epigenomics, metabolomics, and methylomics. Microbiota- and gut health-based approaches included microbiota sequencing methods, microbiota-directed complementary foods and therapies, and the use of anti-inflammatory nutritional approaches to modulate the microbiome. Devices, apps, and digital health included non-invasive and minimally invasive tools, such as wearable devices, point-of-care devices, application-based tools, digital health diagnostics, artificial intelligence, and other portable technologies.

The experts provided insights into the barriers that may impede the implementation of these precision nutrition approaches. As previously, we analyzed the barriers to implementation discussed by experts into four themes: resources (76.9%), awareness (46.2%), systems level (38.5%), and collaboration (30.8%) barriers to implementation of precision nutrition in LMIC (Figure 5B). Notably, resources emerged as the top barrier to implementation, followed by awareness, which included an adequate amount of trained scientists and skilled workforce (Table 5).

### 3.3. Additional Considerations for Precision Nutrition Implementation in LMIC

The opinion leaders and a wider group of experts who attended the SFNV workshop collaboratively discussed key challenges and opportunities for innovations in one of two precision nutrition topics: (1) the collection and analysis of nutritional status and dietary intake data, or (2) nutritional products, supplements, and treatments to improve nutritional status and health in LMIC populations (Table 6).

#### 3.3.1. Considerations for Precision Nutrition to Improve the Collection of Dietary Intake and Nutritional Status Data in LMIC

The workshop attendees emphasized the lack of equitably deployed, precise, and affordable tools in LMIC. Consequently, nutrient intake and nutritional status data are sparse. In addition, the precision nutrition concept for LMIC must be distilled down into simple communications and impactful applications. Investments in both dietary assessment methods and reliable nutrition composition tables and analyses are needed. While newer methods such as mobile food records and image capture [13,49,50] have been used in certain contexts, the capacity for data analysis is lacking in many LMIC. A pipeline is needed to specify the needs for accurate collection and analysis of samples and to ensure that information collected is managed under ethical and data improbability standards [51,52]. More research and investments into identifying biological mechanisms and validating existing and new candidate nutrient biomarkers and tools are essential for progress in precision nutrition applications. Precision nutrition applications were considered promising for LMIC, if they can be scaled up sustainably and meet regulatory guidelines (Table 6).

#### 3.3.2. Considerations for Precision Nutrition to Improve the Design of Nutritional Products, Supplements, and Treatments

Some workshop experts suggested that the end goal of precision nutrition is adequate nutrient intake and status through diets across populations—stratified to the specific target groups, when necessary—to reach nutritional requirements. Other experts argued that stratification of interventions is needed, as there is currently a strong focus on one-size-fits-all approaches in public health nutrition that can be harmful to certain groups in certain contexts, as observed in the Pemba trial [28,29,30]. Safety and “do no harm” [53] (p. 300) were considered essential components to the design of precision nutrition-based interventions for LMIC.

The workshop experts suggested that social marketing strategies to build knowledge, drive behavior change, and incorporate healthy practices into consumers’ lives should be part of the nutritional solutions [54,55]. They recognized that changing diets is not an easy task due to affordability, accessibility, among other challenges; therefore, targeted supplementation and fortification efforts may be needed. The experts expressed that precision nutrition must be built around these key elements and goals as well as safety, ethics, and impact. Furthermore, prioritizing locally developed solutions was considered essential due to cultural differences and to support local economies [55]. Precision nutrition was viewed as useful for discovery science and to enable the evolution from one-size-fits-all approaches to accurate, targeted approaches. The diverse discussion points of the brainstorming session outlined in Table 6 demonstrate the need for more accurate information when designing nutritional solutions to ensure that they will have the desired impact.

### 3.4. Collaboration to Accelerate Precision Nutrition for LMIC Is Supported by Local Nutrition Experts

As part of the expert opinion survey and workshop, interest in further collaboration with the SFNV to support the implementation of precision nutrition for LMIC was discussed. All experts of the opinion survey were open to this further collaboration, and approximately one third of the opinion leaders viewed collaboration as an essential element for implementation (Figure 5B). There was also interest in collaborating toward proofs-of-concept by the wider group of experts who attended the workshop.

## 4. Discussion

The current global food and nutrition challenges require bold action to put an end to malnutrition [24] before further population expansion [56] and climate change [57] cause additional stresses on the system and exacerbate nutritional challenges. The emerging field of precision nutrition represents an optimistic path forward in public health nutrition in part by filling data gaps needed to design more efficacious nutritional interventions and to reduce unintended harm [28,29].

The SFNV Precision Nutrition for LMIC project aims to make the use of precision nutrition approaches more accessible, affordable, and desirable to LMIC to address the most pressing nutrition challenges of today. Because precision nutrition has remained primarily confined to affluent individuals in high-resource settings, it was not yet known whether precision nutrition was considered a relevant approach to tackle malnutrition in LMIC. A more thorough understanding of region-specific considerations was needed.

We investigated the food, nutrition, and related health challenges facing a range of populations. The barriers and current approaches to addressing these challenges and precision nutrition opportunities were analyzed to determine the relevance of precision nutrition as part of the solution. In this expert opinion survey, the working group reports qualitative and quantitative findings from nutrition experts who live or work in LMIC. Overall, this study revealed that tailored nutritional approaches with enhanced accuracy and specificity can find further relevance for target groups in low-resource settings; however, awareness, resources, collaboration, and systems level barriers must be considered and addressed for successful implementation of sophisticated precision nutrition methods in their regions.

When further discussing the relevance of precision nutrition for LMIC during the workshop, the working group found wide support for its relevance for at-risk target groups. In addition, safety, ethics, impact, and affordability surfaced as key priorities for implementation. The workshop also revealed that further efforts are required to clarify the concept of precision nutrition, especially for implementation in low-resource settings.

The term ‘precision nutrition’ lacks clarity in the literature [1,2,3,4,5,6,7,8,9,10,11,12], causing multiple interpretations surrounding its meaning in different contexts. The working group believes precision nutrition for LMIC follows a public health nutrition process with an emphasis on the targeting of population subgroups to achieve impactful solutions for all those in need, not just for those who respond positively to an intervention. 

Through this study, we propose a new term “precision public health nutrition”, to help clarify this concept further and improve the understanding of the SFNV Precision Nutrition for LMIC projects’ mission. The working group defines precision public health nutrition as a data-driven process to accurately define nutrition and related health challenges and current solutions, to stratify at-risk target populations according to biological, ecological, social, behavioral, and environmental factors, in order to develop tailored, evidence-based, accessible, affordable, desirable, safe, and well-informed nutritional recommendations and solutions that offer measurable benefits.

The valuable insights exchanged with the expert opinion leaders provided a deeper understanding of the need for more precision in public health nutrition to improve elements of safety, ethics, and impact for all at-risk subgroups. As a result, the working group developed a seven-step process of precision public health nutrition, leading to context-dependent, safe, ethical, and impactful nutritional recommendations (Figure 6). Precision public health nutrition takes a public health nutrition approach but takes it a step further by applying precision to the solutions and recommendations through diverse methods of data collection and analysis. This dynamic investigation and the stratification of target groups can support more subgroups and reduce unintended harm. The integrative methods promise to offer greater accuracy, efficiency, and impact in LMIC compared to the current nutritional intervention.

First, nutritional challenges and target populations must be identified, such as through a needs assessment with local leaders. Second, current methods and solutions of the target groups must also be analyzed to effectively determine gaps and needs. Third, performing a range of precise approaches to collect and analyze detailed data on biological, ecological, social, behavioral, and environmental factors must be performed to gather accurate information that is context dependent.

Upon analysis of the comprehensive data, the fourth step of the precision public health nutrition process is to identify and develop tailored nutritional solutions to mitigate corresponding challenges in the target group, and subgroups when needed. Fifth, tailored interventions for subgroups are implemented. Sixth, the intervention data are collected and analyzed for the development of well-informed, evidence-based nutritional recommendations, as depicted in Step 7, or the process cycles back to Step 1. The cycle will repeat to ensure precise, effective solutions are implemented for the nutritional challenges and subgroups in need. Importantly, precision public health nutrition efforts must prioritize safety, ethics, and impact.

It is well known that food and nutrition issues and related health challenges exist in LMIC [24]. Our study identified specific challenges in LMIC regions and subpopulations within each region that can be in part mitigated with precision public health nutrition efforts. We noted some overlapping nutritional challenges, such as micronutrient deficiencies, anemia, stunting and wasting, poor gut health, and non-communicable diseases like hypertension. Diets rich in plant-based foods that consist of reduced calorie and cholesterol intake have been shown to prevent and mitigate malnutrition and noncommunicable diseases [58,59,60]. However, context remains a key element for determining dietary recommendations. For example, regional accessibility, supply chains, consumer preferences, and food safety impact consumption patterns of plant-based food sources in specific regions [61]. A context-specific approach to nutritional recommendations may identify the need for solutions that include fortification or demand generation in a particular region or for a particular subgroup [62].

Reaching the most people through one-size-fits-all approaches inherently hinders the precision and accuracy of interventions and recommendations for some subgroups. Implementing tailored evidence-based approaches with improved specificity, safety, and efficacy has potential to yield impactful outcomes among at-risk subpopulations. However, stewardship, capacity and training, resources, and funding are needed for precision public health nutrition opportunities to be feasible for LMIC. 

Due to the lack of adequate data collection and management systems, nutrition efforts must aim to expand field-feasible and validated technologies and accessible information technology platforms. Then evidence-supported, safe, tailored solutions for subgroups must be implemented in a sustainable manner. Point-of-care devices may be a valuable, easy-to-use tool for identifying micronutrient deficiencies in LMIC, thereby facilitating wider application in the clinic and field settings with minimal invasiveness [31]. Yet, it is important for these tools to become affordable and undergo validation through randomized control trials to assess efficacy and accuracy compared to standard methods. Furthermore, logistical challenges and uncertainties in the implementation process of point-of-care tools and other precision nutrition approaches remain.

Our analysis warrants additional critique and insights by a wider network of public health nutrition and medical professionals, as well as implementation scientists who conduct nutritional surveys on the ground. It is important that we gather additional insights to better characterize the feasibility of precision public health nutrition for LMIC and to better understand and anticipate implementation challenges at local and national levels. Specifically, additional formative research will be useful to help fill remaining gaps and answer questions, such as:How can the SFNV Precision Nutrition for LMIC project working group best support the nutritional needs of at-risk subgroups through precision public health nutrition approaches?What are the most helpful methods for sampling, handling, and storage of samples for measuring nutrient deficiencies?What aspects of current methods typically used in nutrition and health surveys in LMIC pose challenges and how should existing technologies to measure nutritional status be further innovated to mitigate these challenges?Would use of portable, point-of-care, and/or minimally invasive sampling tools for data collection and analysis be a viable method or pose more logistical challenges if implemented in a survey?How can the implementation of precision public health nutrition solutions be maintained in the long-term?

This study benefited from a diverse panel of experts, many of whom live and work in LMIC; however, limitations in sample size and representation exist. The study could have further benefited from inclusion of additional experts in the field of precision nutrition and representation from more LMIC, such as Nigeria or countries in Latin America with a larger population than other regions represented in this study. Furthermore, because precision nutrition was mostly confined to HIC applications in the literature, the working group utilized their network to identify LMIC nutrition leaders for this opinion survey. Several of the experts held positions in academia, which may have created a bias toward an academic perspective. Future studies must include a broader range of region-specific LMIC professionals in public health nutrition and medicine defined systematically through database searching. In addition, an assessment of baseline knowledge of precision nutrition followed by training opportunities and a post-intervention assessment would facilitate a more robust statistical analysis for measuring support and potential of precision nutrition opportunities in various regions.

This study was the first to assess the relevance of precision nutrition for LMIC and describe the support and barriers for its applications in low-resource settings by key experts. LMIC nutrition leaders supported collaboration to work toward implementation of precision nutrition approaches to mitigate nutritional challenges in their regions. The overarching goal of precision nutrition is to help improve the adequate intake of nutritious foods and nutritional status through evidence-based recommendations, and to maintain good health across the life stages for all. Tackling the nutrition challenges of today requires accurate data and a deep understanding of the needs and challenges in local contexts. Failure to expand precision nutrition opportunities and benefits to LMIC populations threatens to further widen the health disparities between high- and low-resource regions and to limit research and interventions needed to support the nutrition and health of diverse groups. Precision public health nutrition will more often have application in LMIC at the subpopulation level to help mitigate nutritional challenges with elements of safety, ethics, and impact at the center of the approach.

## 5. Conclusions

Precision nutrition is a dynamic process that gathers context-specific data and translates it into evidence-based recommendations and solutions to improve human health and nutrition. While precision nutrition is typically implemented at the individual level in high-resource settings, the process of precision nutrition also supports the targeting of at-risk groups at the subpopulation level in LMIC. The findings of this study indicate that precision public health nutrition, an emerging concept from the SFNV Precision Nutrition for LMIC project working group efforts, is relevant and of interest to address nutritional challenges in at-risk target groups, and therefore, may serve as a promising way forward to support local and global efforts to improve nutrition and health safely and effectively in low-resource settings.

## Figures and Tables

**Figure 1 nutrients-15-03247-f001:**
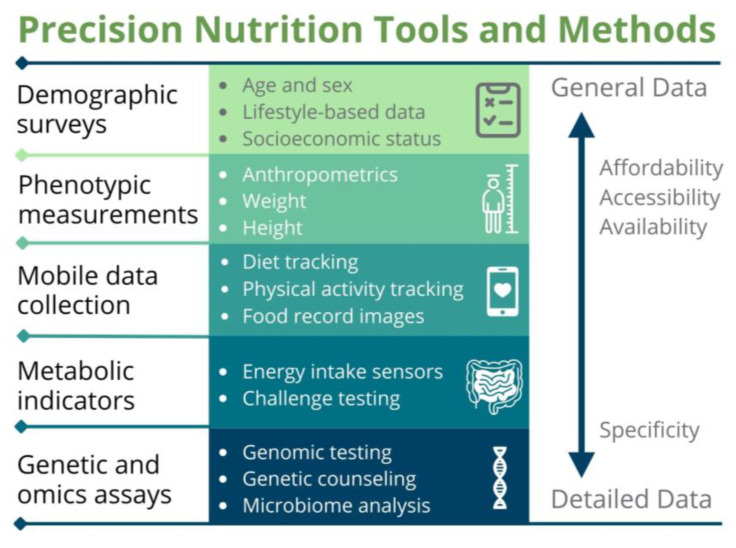
Precision nutrition includes several approaches that range in specificity, affordability, accessibility, and availability. Certain precision nutrition approaches yield detailed data from methods with high specificity (bottom), while others provide general, less detailed data from methods of wider affordability, accessibility, and availability, but less specificity (top). Detailed and general data from the specific and wider methods, respectively, contribute to the development of well-informed, tailored nutritional recommendations and solutions. Adapted from Adams et al., 2020 [1] and Gavin-Smith et al. 2022 [7].

**Figure 2 nutrients-15-03247-f002:**
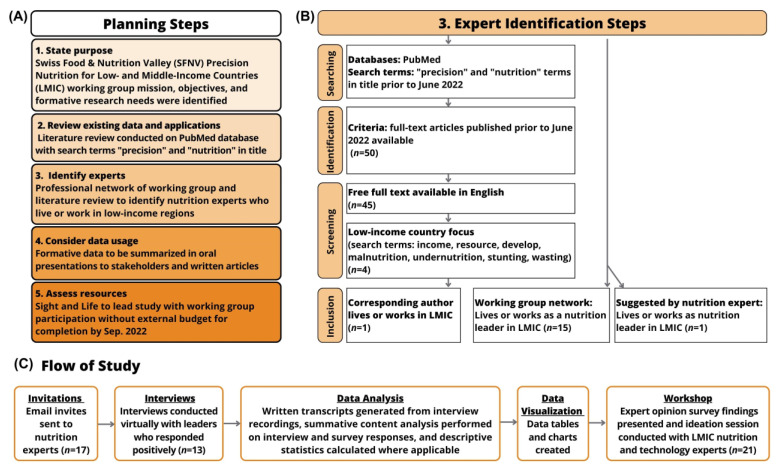
The planning procedure for the expert opinion survey. (**A**) The planning steps include stating the purpose, reviewing existing precision nutrition data and applications, identifying experts, considering usage of data, and assessing available resources. The Swiss Food & Nutrition Valley (SFNV) project working group defined the mission, objectives, and formative research needs for the Precision Nutrition for low- and middle-income countries (LMIC) project. A literature review was conducted in PubMed for full text articles containing the terms “precision” and “nutrition” in the title. Experts in nutrition in LMIC were identified and invited for participation in the study. Data would be used as formative research presented orally and as written articles. Available resources for conducting the study including the project lead, time for completion, and budget were assessed. (**B**) Expert identification steps include a literature review of free-full text, English articles with a low-income country focus, and the corresponding author is a nutrition expert living or working in LMIC. The working group listed colleagues from their network, and some were suggested by the LMIC nutrition experts. (**C**) The study design flow: invitations sent to experts, interviews conducted, data analysis and visualization, and a workshop hosted by the working group to disseminate key findings.

**Figure 3 nutrients-15-03247-f003:**
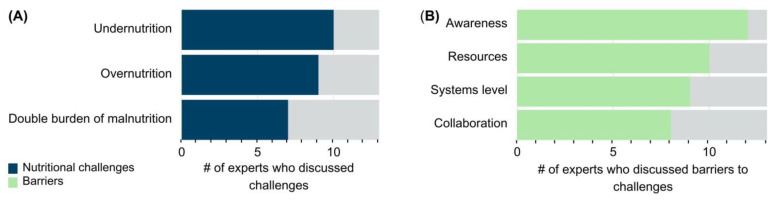
Reported nutritional challenges in LMIC and barriers to mitigating these challenges. (**A**) Number (#) of experts who mentioned undernutrition (including micronutrient deficiencies), overnutrition, or the co-existence of under- and overnutrition called the double burden of malnutrition as the top nutritional challenges in LMIC. (**B**) Number (#) of experts who mentioned barriers to addressing the nutritional challenges analyzed into four themes: awareness, resources, systems level, and collaboration barriers.

**Figure 4 nutrients-15-03247-f004:**
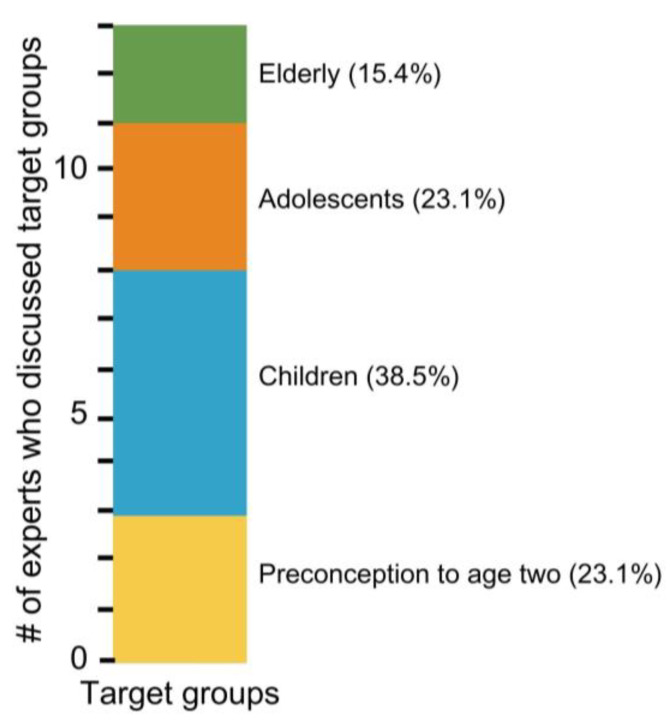
Number (#) of experts who chose a top target group for intervention to mitigate nutritional challenges. Groups include elderly (green), adolescents (orange), children (blue), and preconception to age two (yellow). The percentage (%) of experts who chose the target group is included.

**Figure 5 nutrients-15-03247-f005:**
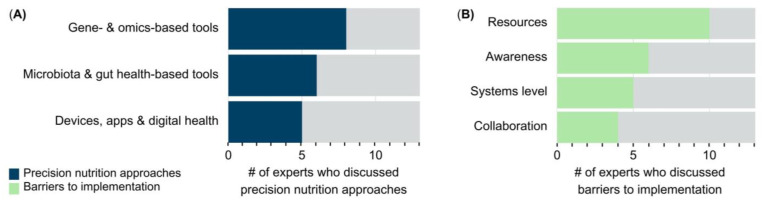
The top precision nutrition methods to tackle nutrition challenges and the barriers to implementation in LMIC. (**A**) Number (#) of experts who listed the top precision nutrition approaches that could help mitigate nutritional challenges in LMIC. (**B**) Number (#) of experts who mentioned barriers to implementing precision nutrition approaches in themes of awareness, resources, systems level, and collaboration barriers.

**Figure 6 nutrients-15-03247-f006:**
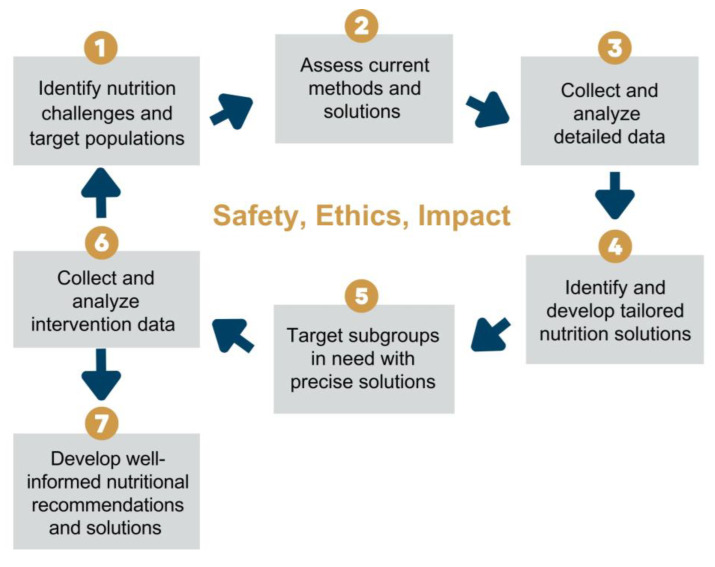
The seven-step process of precision public health nutrition has safety, ethics, and impact at its core. Step 1: nutritional challenges and target populations are identified. Step 2: current methods and solutions are assessed. Step 3: precise approaches to collect and analyze detailed data are performed. Step 4: tailored nutritional solutions are identified and developed to mitigate the nutritional challenge. Step 5: subgroups in need are targeted with precise solutions. Step 6: the intervention data are collected and analyzed to develop well-informed nutritional recommendations and solutions (step 7), or the process cycles back to step 1 to repeat and ensure precise, effective approaches are implemented to address the nutritional challenges of target groups.

**Table 1 nutrients-15-03247-t001:** Food, nutrition, and related health challenges in LMIC reported by the opinion leaders.

Food and Nutrition Challenges	Related Health Challenges
Poverty	Underweight
Food security	Stunting
Food affordability	Wasting
Food production system	Anemia
Food policy	Acute illness
Undernutrition	Low anthropometry
Protein-energy malnutrition	Failure to thrive
Micronutrient deficiencies/hidden hunger	Mortality
Overnutrition	Overweight
Double/triple burden of malnutrition	Obesity
	Hypertension
	Diabetes
	Dyslipidemia
	Steatosis
	Metabolic Syndrome
	Cardiovascular disease

**Table 2 nutrients-15-03247-t002:** Reported barriers to addressing food and nutrition challenges by theme.

Awareness	Resources	Systems Level	Collaboration
Nutrition education	Tools and technologies	Poverty	Coordination
Media and other influence	Data availability & management	Food security	Organization
Trained and engaged workforce	Funding	Budget	Engagement
Communication of guidelines	Intervention coverage	Locally developed solutions	Human resource management
Ethics	Sanitation	Organization of systems	Other collaboration
Habit	Other resources	Accessibility and availability	
Conditioning		Policies and guidelines	

**Table 3 nutrients-15-03247-t003:** Critiques of the working SFNV definition of precision nutrition.

Critique Type	Critique
Term	
‘Individual’	Too many population-wide problems to focus on the individual Can be focused on the community or regional level Needs to be a gradation through stratification of groups with similar characteristics to individual precision nutrition Can do a one-size-fits-all approach in areas where there is little precision nutrition happening, and then leverage funding and commitment to do stratified targeting
‘Rigorous’	Rigorous is not necessary or well-defined for low-resource settings
‘LMIC’	Precision public health is for all Precision nutrition is global, not just for LMIC
‘Accessibility and affordability’	Consider if accessibility and affordability are obligatory
2. Definition-wide	Should take the ecology of a person into account Needs to be broad, but also needs depth Too broad and not precise

**Table 4 nutrients-15-03247-t004:** Summary of current approaches to address the nutrition challenges described in the expert opinion survey.

Public Health Approaches	Research	Data Resources
Interviews and surveys	Etiology of nutritional challenges	World Health Organization
Awareness campaigns	Biology of phenotypes and deep phenotyping	United Nations Children’s Fund
Education sessions and trainings	Clinical markers and biomarkers	Food and Agriculture Organization of the United Nation
Anthropometry	Bioavailability of micronutrients in foods	World Food Programme
Blood biomarker biochemistry	Toxins	Governments
Point-of-care measurements and technology (e.g., blood glucose monitors)	Home gardening and dietary diversity	Other non-government organizations
Food-based interventions	Complementary foods	Ministry
Whole-value chain approaches	Implementation research	

**Table 5 nutrients-15-03247-t005:** Reported barriers to implementing precision nutrition approaches in LMIC.

Awareness	Resources	Systems Level	Collaboration
Precision nutrition knowledge	Funding	Affordability	Capacity
Skilled workforce	Suitable technology	Acceptance	Collaborative efforts
Trained scientists	Minimally invasive tools	Data privacy and standardization	Collaborative research
Consumer awareness	Research and data	Government regulation and guidelines	Human resource management
Ethics	Resource allocation	Data protection laws	
	Health records	Food production system	
		Universal health coverage and health systems	
		Commercialization focus	

**Table 6 nutrients-15-03247-t006:** SFNV Precision Nutrition for LMIC Virtual Workshop topics and points of discussion.

**1. Precision Nutrition—the Collection of Dietary Intake and Nutritional Status Data**	
**Topic**	**Discussion points**
Key nutrition and data challenges	Accuracy of current methods Accessible and affordable technology such as point-of-care and omics tools Discovery and validation of approaches Safety and care of technologies Coding and data collection, analysis, and availability Sustainable business models for scaling up Environmental concerns
Opportunities in precision nutrition to mitigate the challenges	Creation of an open-access food composition database for use across countries Harmonization and standardization of methodology for dietary intake measurements Cameras, mobile, and other digital solutions for dietary intake data analysis Point-of-care diagnostics and noninvasive tools that are affordable and for nutrition by design Digital health records Facilitation of sample collection and data analysis in one platform to provide a centralized data set Omics methods: genomics, proteomics, lipidomics, urinary metabolomics
Suggested immediate next steps	Define clear objectives, technologies, and devices, and create needed specifications including standardization of collection methods Move toward intervention trials using decentralized labs Use local and affordable solutions based on the local context Support local training, equipment transfer, and advocacy Leverage already existing medical channels like viral testing mechanisms in rural health care settings to implement approaches
**2. Precision Nutrition—Nutritional Products, Supplements, and Treatments**	
**Topic**	**Discussion points**
Current nutritional treatments or interventions for nutritional challenges	Food, diet, complementary food, (e.g., eggs for undernutrition [43,44], *Moringa oleifera* in maize for iron deficiency [45,46]) Dietary restriction for issues such as Celiac disease Ready-to-use supplementary or therapeutic food (RUSF or RUTF) Iron-folic acid supplementation (IFA) and multiple micronutrient supplementation (MMS) Vitamin A supplementation
Precision nutrition approaches to mitigate the nutrition challenges	Point-of-care diagnostics for anemia and micronutrient deficiency Bouillon cubes with phytase for protein deficiency Biomarkers and assays for Environmental Enteric Dysfunction Microbiota-directed complementary food for acute malnutrition [47,48] Re-examining IFA in at-risk groups and intervention stratification
Key steps to turn approaches into solutions and pitfalls to avoid	Recognize that precision nutrition can be preventive or curative Develop local nutrition solutions using local foods where possible Compare precision nutrition approaches with current approaches Assess safety of current solutions Identify causes of the nutritional challenges for target groups Address awareness, affordability, and availability Consider social marketing for demand generation and education Assess regulatory, implementation, and logistical needs for scaling up Consider resilience of crops as nutritional products Keep safety, ethics, and impact at the forefront

## Data Availability

Data is available within the article and Appendix A. The data not presented in this study are available on request from the corresponding author. Certain data are not publicly available due to privacy restrictions.

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
