# Peer review of "Precision Nutrition Opportunities to Help Mitigate Nutrition and Health Challenges in Low- and Middle-Income Countries: An Expert Opinion Survey"

_nutrients, 2023, doi:10.3390/nu15143247_

Round 1

Reviewer 1 Report

Reviewer comments

In this work, Bedsaul et al. presented findings from expert engagement regarding the region-specific nutritional challenges; current methods and solutions; precision nutrition relevance and opportunities; collaboration; considerations for developing projects; and the way forward from the research and implementation perspectives in LMICs. Kindly, find below my comments for your response.

Introduction:

The Introduction is generally well-written.

Materials and Method:

The authors should kindly state the search terms that were adopted for the search process. They could consider listing the databases that were searched as well. The depth of the work could have been greatly enhanced if the authors conducted a scoping review with Bibliometric analysis of the literature. This visually could have revealed where the experts engaged in the “precision nutrition space” are affiliated and their past research work.

What informed the sample size of the participants used for the study? This is important to ensure that truly the sample size was representative and statistical inferences could be made from that.

Line 240-242: It is surprising that Nigeria was not represented in the countries listed. Could the authors explain why? Nigeria has a higher population and quite a heterogenous population in terms of tribal orientation. Yet, it was not listed. Meanwhile, at the acknowledgement section, I see two Ghanaians (Mrs. Audrey Essilfie and Dr. Matilda Steiner-Asiedu) included. Ghana has about 32 million population, which is way lower than Nigeria which has about 200+ million. I think that could be a limitation. To ensure homogenous representation, one individual from Ghana and another from Nigeria would have been a good idea.

Data analysis: The authors indicated Excel was used for the data analysis. Was that just for the descriptive statistics? For such a topic, perhaps, in the future, the authors could conduct a baseline assessment of participants (end-users) understanding of the use of “precision nutrition” and post-intervention assessment of their understanding after training. A T-test analysis could then be used to determine whether significant differences in mean existed. The authors could have also used PCA to discriminate between the participants responses in this present study. Presently, the work reads more like a report.

Results

The information presented at Lines 231-236 should come under “Discussion” and not under the “Results section”.

Reviewer 2 Report

Dear authors,

the topic of your manuscript is interesting and valuable. However, you should reconsider the title. Please, make the title more clear.

I have also some other remarks:

1. Figure 1. It has been already published. Make our own figure, which represents this information.

2. Section. 2.1.1. Planning:

Please provide a graphique which explains the planning procedure.

3. I have detected some spelling mistakes. I recommend your manuscript to undergo English editing. You can use MDPI English editing services or other option.

4. Figure 3. Please, provide a better quality figure.

5. Table 3. It is unclear. Revise it and make it more user-friendly.

6. You have included boxes in the manuscript. Remove the boxes and include this information in the main text.

7. The manuscript is not fully organised according to MDPI guidelines. Check the manuscript.

8. References: You could include some other valuable references which refer to nutrition and general health:

 https://doi.org/10.3390/foods10123052

https://doi.org/10.3390/nu9080848

https://doi.org/10.3390/nu11112712

I recommend the manuscript to undergo English editing. There are some minor mistakes.

Round 2

Reviewer 1 Report

I am happy with the revisions made. 

Author Response

Dear Reviewer 1,

We are glad to hear you are happy with the revisions made to our article.

Thank you again for the helpful suggestions.

Sincerely,

Jacquelyn Bedsaul, on behalf of all co-authors

Reviewer 2 Report

Dear authors,

the quality of your manuscript has been significantly improved.

Author Response

Dear Reviewer 2,

We are glad to hear that the revisions we made significantly improved our article.

Thank you again for the helpful suggestions and feedback.

Sincerely,

Jacquelyn Bedsaul, on behalf of all co-authors